# Antibacterial Activity Potential of Industrial Food Production Waste Extracts against Pathogenic Bacteria: Comparative Analysis and Characterization

**DOI:** 10.3390/foods13121902

**Published:** 2024-06-17

**Authors:** James Ziemah, Matthias S. Ullrich, Nikolai Kuhnert

**Affiliations:** School of Science, Constructor University Bremen, 28759 Bremen, Germany; jziemah@constructor.university (J.Z.); mullrich@constructor.university (M.S.U.)

**Keywords:** phytochemicals, coffee silverskin, lemon peel, hot trub, α-amylase inhibition, antibacterial, polyphenols

## Abstract

The Food and Agricultural Organization estimates a 17% loss in the food production chain, making it imperative to adopt scientific and technological approaches to address this issue for sustainability. Industrial food production waste and its value-added applications, particularly in relation to a wide variety of pathogenic microorganisms and the health-related effects have not been thoroughly investigated. This study explores the potential of food production waste extracts—lemon peel (LP), hot trub (HT), and coffee silverskin (CSS) as sources of bioactive compounds. Extraction was conducted using hydro-methanolic extraction with yields in LP (482 mg/1 g) > HT (332 mg/1 g) > CSS (20 mg/1 g). The agar diffusion assay revealed the substantial antibacterial activity of all three extracts against *Erwinia Amylovora*, *Escherichia coli*, *Bacillus subtilis*, and *Bacillus aquimaris*. All extracts demonstrated activity against Gram-positive and Gram-negative bacteria, displaying minimum inhibitory concentrations effective against pathogenic bacteria like *Listeria monocytogenes*, *Staphylococcus aureus*, *Vibrio parahaemolyticus*, and *Salmonella enterica*. Total phenolic content (TPC in mg GAE/1g) was 100, 20, and 100 for CSS, HT, and LP, respectively. Antioxidant activity by ABTS indicated IC_50_ of 3.09, 13.09, and 2.61 for LP, HT, and CSS, respectively. Also, the antioxidant activity of the extracts was further confirmed by DPPH assay with the best activity in CSS (9.84 GAEg^−1^) and LP (9.77 mg of GAEg^−1^) rather than in HT (1.45 GAEg^−1^). No adverse cytotoxic effects on HaCaT cells were observed. Pancreatic amylase inhibition demonstrated antidiabetic potential, with LP showing the highest levels (92%). LC-MS characterization identified polyphenols as the main compounds in CSS, prenylated compounds in HT, and flavanols in LP. The findings imply the potential sustainable use of food production waste in industry.

## 1. Introduction

Globally, 1.3 billion tons of food is wasted annually, translating into 940 billion US dollars and contributing 10% of greenhouse gases. Food production waste, generated across the supply chain, comprises agricultural, industrial, and distribution processes that have a global environmental and economic impact. Household activities contribute 42%, and the food industry and services sectors contribute 39% and 14%, respectively, with 5% lost during distribution [1,2]. Disposal of food production waste in landfills releases methane, contributing to climate change. Extracting bioactive compounds, fibers, and proteins from food production waste is crucial for addressing environmental and economic impacts. The sustainable recovery of bioactive compounds from plant waste is essential for enhancing food availability and reducing reliance on cash crops, contributing to industrial food sustainability [3].

Polyphenols, alkaloids, terpenoids, and sulfur-containing compounds in plant secondary metabolites evolved as defense mechanisms during plant growth [4]. Polyphenols, specifically, can combat chronic diseases related to oxidative stress while the cytotoxicity of plant extracts is assessed using specific cell lines [5]. Several bioactive compounds found in plants such as chlorogenic acid, caffeic acid, genistein, and scopoline, which have potential applications in drug delivery, food additives, and antibacterial agents, are discarded in food production waste [6].

Combinatorial extraction methods, such as solid–liquid and ultrasonic-assisted extraction, are efficient in breaking down plant cell walls to release bioactive compounds. The combination of extraction methods ensures high yields of phytoextracts. Coffee silverskin, rich in polyphenolic compounds, and hot trub, containing α- and β-acids exhibit excellent antibacterial activities. Lemon peel, with flavonoid glycosides and other compounds, demonstrates antibacterial and antifungal activities [7]. Comparative study of the development of natural antimicrobial agents from industries as alternatives to synthetic chemicals is highly imperative as these are safer. Boosting the food production industry’s sustainability by recovering the valued added compounds is essential for the future food, cosmetics, hygiene, and nutritional health of consumers [3,8]. The phytochemicals with relevant value-added applications in CSS are Ferulic acid (38.69–145.32 µg g^−1^), Rutin (1.25–10.65 µg g^−1^), Quercetin (3.06–3.76 µg g^−1^), Kaempferol-3-glucoside (0.36–1.89 µg g^−1^), and Hyperoside (0.32–0.87 µg g^−1^) [9]. From 8 tons of roasted coffee, about 60 kg of CSS is generated [10]. The amount of hot trub generated per annum is 250 thousand tons [11]. Hot trub contains important phytochemicals in high amounts such as xanthohumol (239 μg g^−1^), Iso-α-acids (6 mg g^−1^), and α-acids (3.6 mg g^−1^) [12]. Similarly, lemon waste production accounts for about 10 million tons annually, which is 50–70% of the raw fruit [13]. Major compounds known in lemon peel extract are Orientin (1104.39 ± 46.76 µg g^−1^), Eriocitrin (1041.17 ± 29.64 µg g^−1^), Apigenin 6,8-di-C-hexoside (608.96 ± 22.16 µg g^−1^), Luteolin (971.16 ± 55.31 µg g^−1^), Naringenin-7-glucoside (303.04 ± 22.28 µg g^−1^), Rutin (547.65 ± 12.87 µg g^−1^), and many more. The presence of these known compounds gives lemon peel extract several biological usages including anti-inflammatory, antimicrobial activity, neuroprotective, cardioprotective, and other pharmacological application [14,15].

The development of antibiotic resistance in microorganisms has necessitated the current interest in plant-based natural products for eliminating pathogenic microorganisms. This work aims to compare the extraction yield of bioactive compounds from three industrial food production wastes. Using various antioxidant assays, the most active waste extract will be identified, and enzyme inhibition activity will be tested. Antibacterial activity will be ascertained against both pathogenic and non-pathogenic organisms. Finally, the compounds in the extracts will be characterized using ultra-high-performance liquid chromatography–electrospray ionization–quadrupole-time-of-flight mass spectrometry (UHPLC-ESI-QTOF-MS).

## 2. Materials and Methods

### 2.1. Chemicals and Reagents

The chemicals used in the experimental pipeline included: methanol LC-MS grade and formic acid for mobile phase modifier and calibration buffers as HPLC grade (both purchased from Carl Roth, Karlsruhe, Germany), methanol of synthesis for extraction (Purity ≥99% synthesis grade purchased from Carl Roth, Karlsruhe Germany); Acetonitrile was used for the HPLC sequence measurement (Acetonitrile for UHPLC gradient, J.T Baker, Gliwise, Poland); Milli-Q water Analyzed LC-MS Reagent (J.T. Baker, Gliwice, Poland); Gallic acid (Sigma-Aldrich, Steinheim, Germany) Acetic acid (PanReac Applicehem, Darmstadt, Germany); and Potato starch (Carl Roth, Karhsruhe, Germany were used for amylase inhibition assay; Dinitrosalicylic acid (DNSA) reagent (Sigma-Aldrich, Steinheim, Germany); Sodium potassium tartrate (Carl Roth, Karlsruhe, Germany); and Sodium hydroxide (Carl Roth, Karlsruhe, Germany). The remainder included Lysogeny broth (LB) media; sterilized agar plates; and sterilized 96-well plates.

### 2.2. Food Production Waste 

Coffee silverkin (CSS) was generously donated by Gollücke & Rothfos GmbH Bremen, Germany, for the testing of waste management and its reuse for innovative applications. CSS consists of a thin envelope directly attached to a coffee bean. During roasting, coffee silverskin is recovered after moisture loss [8]. Hot trub (HT) from the Bremen beer brewery company, Germany (Becks GmbH & Co., KG, AnBev, Bremen, Germany), was used for analysis of bioactive compounds. HT is a beer slurry waste generated in the brewery industry during beer production. It is the most abundant waste in the industry after spent grain waste. The most common components of hot trub are the hop particles, unstable colloidal protein particles, and carbohydrates. HT is composed of raw material waste from barley, hops, and other sources [16]. Lemon peel (LP) is waste generated in the beverage industry and was purchased from Henrich Klenk GmbH and Co. KG (Schwebheim, Bavaria, Germany). LP contains several bioactive compounds such as polyphenols and lipids, making it a potential recyclable material for industrial dietary food applications [17].

### 2.3. Extraction Procedure

The crude phytoextracts from the three food production wastes were assessed by the ultrasonication and maceration methods simultaneously in a stepwise manner while agitating the mixture with a magnetic stirrer as previously reported [18] with modifications. Briefly, 1 g of food production waste was weighed and 10 mL of 70% MeOH solution was added. The mixture was then sonicated in a water bath for 15 min. The mixture was stirred using a magnetic stirrer for 1 h. The extract was filtered using a filter paper filter over a funnel. The extraction procedure was repeated three times. The extract solvent was evaporated in a rotary evaporator, frozen at −80 °C, and lyophilized into powder. The dried weight was quantified to obtain the crude extract yield. This process was repeated in triplicates, the averages and standard deviations were calculated, and their yield was compared. The obtained phytoextract was stored at −20 °C for further analysis.

### 2.4. Antibacterial Screening of Phytoextract

#### 2.4.1. Agar Diffusion Assay 

The agar diffusion method is well known for determining inhibition zones on agar plates. This method was used for screening the antibacterial activities of the crude phytoextract as described earlier [5], with some modifications. The sterilized LB agar was prepared as previously described [19], 20 mL was poured into each sterilized plate and allowed to solidify in a flow chamber. The LB agar plates were stored at −20 °C until use. Phytoextracts were screened against the following non-pathogenic organisms: *Erwinia amylovora* (EA); *Escherichia coli* (EC), *Bacillus subtilis* (BS), and *Bacillus aquimaris* (BA) using 100 µL of a defined organism. An optical density of 0.1 of the bacteria solution spread over an agar plate using sterile glass beads. Bacteria-covered agar plates were prepared with wells filled with 100 µL of extract solution. Controls included water, 70% MeOH, or ampicillin in the wells. After overnight incubation, zones of inhibition were assessed, reflecting the phytoextract’s bioactivity against microorganisms. The antibacterial activity test used in this experiment explored two solvents: an aqueous solvent and a 70% methanol solvent for dissolving the extracts.

#### 2.4.2. Minimum Inhibitory Concentration (MIC) Assay

The MIC analysis was performed against seven pathogenic organisms: *Listeria monocytogenes*, *Vibrio parahaemolyticus*, *Chronobacter sakazakii*, *Pseudomonas aeruginosa*, *Salmonella enterica*, *Klebsiella pneumoniae*, and *Staphylococcus aureus*. The serial dilution method was used to determine the MIC of the phytoextract in a sterilized 96-well plate as described previously [19], with modifications. The pathogenic organisms were pre-cultured overnight using a single colony of bacteria. LB broth was used to dilute the pre-culture to attain an O.D. of 0.1 at 600 nm wavelength, measured using a spectrophotometer. The O.D. of 0.1 was further diluted to achieve 1 × 10^5^ bacterial cells/well before analysis. Phytoextracts at known concentrations were added to a series of dilutions of the microorganisms mentioned above. The microorganisms had an approximate concentration of 1 × 10^5^ CFU/mL and were placed in the growth medium in the wells. Each well contained 200 µL of the mixture and the experiments were performed in duplicate. The mixture was then incubated overnight. Microbial growth was observed after the overnight incubation. Phytoextract MICs were compared to experimental controls (ampicillin, only, media, and only water/70% MeOH) performed simultaneously with phytoextracts. The minimum inhibitory concentration of each phytoextract was defined as the minimum concentration at which no bacterial growth was observed. Resazurin is a fluorometric indicator, whose color change aids in determining the MICs of colored extracts by staining [20]. Rezasurin dye solution was prepared at a concentration of 0.2 mg/mL for bacterial staining to determine its MIC. After overnight incubation of the bacterial and extract mixtures on MIC plates, 50 µL of the prepared resazurin solution was added. Ampicillin and distilled water were used as controls and the treatments were observed after 3 h for color changes. Suspensions with surviving bacteria had a color like that of the control (media only).

### 2.5. Lactate Dehydrogenase (LDH) Keratinocyte Cytotoxicity Assay

The human immortalized and non-tumorigenic keratinocyte cell line (HaCaT) was kindly supplied from Prof. Dr. Klaudia Brix’s laboratory (Constructor University). The cytotoxicity of the phytoextracts was tested using a CzQUANTTM LDH Cytotoxicity Assay kit.

### 2.6. UHPLC-ESI-QTOF-MS Characterization

The obtained phytoextracts were characterized using the UHPLC-ESI-QTOF-MS/MS method described before [18] on an Agilent 1260 (Karlsruhe, Germany) UHPLC system in a negative ion mode. The system coupled to a Bruker Daltonics (Bremen, Germany) Impact QTOF MS with an ESI source in the negative ionization mode was used for identifying the secondary metabolites. The Agilent C-18 Poroshell column (0.5 × 250 mm and pore size 2.7 µm) was connected to the guard column (2.1 × 5 mm and pore size 2.7 µm) for the chromatography separation. The mobile phase constituted of 0.01 *v*/*v* formic acid in water (A) and (0.01 *v*/*v* formic acid in acetonitrile (B)). The gradient elution started with 5% B moving up linearly over 25% B in 35 min, this was followed by a 10 min linear increase to 45% B and a 5 min linear increase to 85% B with a total run time of 50 min. To equilibrate the chromatographic conditions before injecting the next sample, each run was followed by a 10 min post run. The samples were injected at a volume of 2 µL and an elution flow rate of 0.350 μL/min. The MS conditions were operated as follows: capillary temperature, 200 °C; drying gas flow rate, 9 L/min; nebulizer pressure, 1.8 bars; funnel 1RF, 300 Vpp; funnel 2 RF, 450 Vpp; hexapole RF of 50, Vpp; in-source CID energy, 0 V; transfer time, 80.0 μs; and pre-pulse storage, 7.0 μs. The eluted compounds were detected in the mass range of *m/z* 80–2000 Da. The tandem mass spectra were obtained in auto-MS^n^ mode (smart fragmentation) by ramping up the collision energy. In the first minute of each chromatographic run, internal calibration was performed using 5 mL of 0.1 M sodium formate solution injected through the six-port valve. The MS data calibration mode used was High precision Calibration (HPC). MS data acquisition was performed using data analysis software 4.2 (Bruker Daltonics, Bremen, Germany). Three processes were employed for the characterization: a literature search of compounds using the Metaboanlyst 6.0 software from BruckerDaltonics, deduction of fragmentation ions from the data, and dereplication. The MS/MS data from the QTOF MS were converted into mzML using proteowizard software version 3.0 (MS convert) and uploaded into an array of software for matching compounds. The extract run underwent matching of the peaks through the Global Natural Products Social Molecular Networking (GNPS), Metlin, and Sirius 5.8.2-win64msi software. The MS data were assigned to compounds in the databases based on high-resolution *m/z* values and tandem MS fragment spectra. The assigned compounds were further confirmed using literature references. Compound assignment relied on the identity within experimental variations in high-resolution *m/z* values and the similarity of the tandem MS fragments ions as described [21].

### 2.7. Spectrophotometric Analysis

#### 2.7.1. Total Phenolic Content (TPC) 

Total phenolic content was determined using a spectrophotometer as described previously [18] with modifications. Briefly, 30 µL of the sample or standard (gallic acid) was mixed with 150 µL of Folin reagent for 5 min at room temperature, and 120 µL of sodium bicarbonate was added. After 90 min of reaction, absorbance was measured using a spectrophotometer at 725 nm.

#### 2.7.2. 2,2′-Azino-bis (3-ethylbenzothiazoline-6-sulfonic Acid (ABTS)) Radical Scavenging Activities

The antioxidant activities of the phytoextracts were measured using a spectrophotometer following previous work [22] with some modifications. Briefly, 10 µL of each sample was serially diluted in a 96-well plate to determine the IC_50_ values of the phytoextracts. The ABTS working solution was prepared and 200 µL was added at a concentration of 0.5 mM. The solution was incubated for 55 min at room temperature and the absorbance was measured at 734 nm of wavelength. IC_50_ was quantified and expressed as mg/mL of gallic acid. The IC_50_ value defines the concentration required by the phytoextract to scavenge 50% of the ABTS initial radicals. The lower the IC_50_ value, the more potent the phytoextract.

#### 2.7.3. 2,2-Diphenyl-1-picrylhydrazyl (DPPH) Radical Scavenging Activity

The DPPH radical scavenging of the extracts was determined as described previously [22] with some changes using a 96-well plate wet chemistry assay. Briefly, 20 µL of Phytoextract solution or gallic acid as a standard was pipetted into a 96-well plate (in triplicate). A DPPH solution of 0.5 mM (200 µL) was added to each well and the mixture was gently shaken. The mixture was incubated in the dark for 30 min and the absorbance was measured at 517 nm.

#### 2.7.4. α-Amylase Enzyme Inhibition Activity

The *α*-amylase inhibitory activity of crude extracts of the food production waste were performed according to the methods described [23], with slight modification. It involved reagents preparation, α-amylase assay optimization, and bioassay of the food waste phytoextract described below.

##### Preparation of α-Amylase Reagents

The Starch solution (1% *w*/*v*) was prepared by boiling 0.5 g of potato starch in 50 mL of milliQ water for 15 min. Dinitrosalicylic acid (DNSA) reagent was prepared by mixing 20 mL of 96 mM DNSA in milliQ water with 5.31 M of sodium potassium tartrate in 2 M of sodium hydroxide (8 mL) and 12 mL of deionized water. 

##### α-Amylase Assay

Experiments were performed to determine the optimum concentration using a concentration series of enzymes ranging from 1 unit/mL–10 units/mL under laboratory conditions. α-amylase (Porcine pancreas, Type VI-B, 23 U/mg of protein) was dissolved in 20 mM of phosphate buffer (pH 6.9) containing 6.7 mM of NaCl to prepare a concentration series. Each enzyme solution (100 µL) was mixed with 100 µL of Milli-Q water in a semi-microcentrifuge tube, incubated at 37 °C for 30 min, and then 100 µL of this mixture was drawn out and mixed with 100 µL of starch solution in another semi-microcentrifuge tube. The samples were incubated again at 37 °C for 10 min and 100 µL of the DNSA reagent was added to the solution and incubated at 85 °C in a heating blot for 15 min. The mixture was diluted with 900 µL of milli-Q water. When preparing blanks for each enzyme concentration, the color reagent solution was added prior to the addition of starch solution to denature the enzyme kept in 85 °C water bath for 15 min, and then diluted with 900 µL of distilled water as explained above. The absorbance of each test solution was measured at 540 nm against the blank and control. Absorbance was determined at each enzyme concentration and the optimum enzyme concentration was determined.

##### α-Amylase Enzyme Bioassay with Food Production Waste Extracts 

Each food production waste phytoextract was dissolved in milli-Q water to obtain a concentration of 50 mg/mL. Each food production extract (100 µL) and 100 µL of enzyme solution with an optimum concentration (8 U/mL) were mixed in a semi-micro centrifuge tube and incubated at 37 °C for 30 min. Subsequently, the same procedure as described above was employed. The same procedure was followed to prepare the blanks, but the DNSA reagent was added before the starch. Negative controls were conducted equivalently by replacing the food production waste extracts with 100 µL of milli-Q water.
(1)$$Percentage\;inhibition\;of\;\alpha \text{-}amylase\;activity=\frac{{Abs}_{negative\;control}-{Abs}_{sample}}{A_{negative\;control}}\times 100\%$$
where, *Abs_negative control_*: absorbance of the negative control at 540 nm; and *Abs_sample_*: absorbance of the sample at 540 nm.

Statistical analysis: The extraction yield, antibacterial activities, antioxidant activity assays, and α-amylase inhibition were performed in triplicate with MICs in duplicate. All the results were presented as means ± SD. One-way analysis of variance (ANOVA) was performed to compare the different food production waste (*p*-value) using OriginPro 2018 64-bit software.

## 3. Results

An initial screening of three food production waste extracts obtained from food production sites in Northern Germany was performed using an antibacterial assay against standard model organisms. The model organisms considered were EA = *Erwinia amylovora* (Gram-negative), EC = *Escherichia coli* (*E. coli*) (Gram-negative), BS = *B. subtilis* (Gram-positive), and *B. aquimaris* (Gram-positive). 

Based on the experimentally observed properties, which included analysis results for solubility and sensory parameters, the HT, CSS, and LP extracts were chosen for further detailed evaluation. The evaluation includes an agar diffusion assay on model bacterial strains, MIC determination including relevant pathogenic bacteria, and chemical characterization of extracts including some parameters of detailed LC-MS profiling.

### 3.1. Antibacterial Activities and Minimum Inhibitory Concentration 

Phytoextracts obtained from three food production wastes were assayed for EA, EC, BS, and BA using various concentrations of lyophilized extract powder (Figure 1). All tested phytoextracts exhibited some level of inhibition against the considered microorganisms. Lyophilized extract powder dissolved at 50 and 100 mg/mL showed good solubility at both concentrations and inhibited the growth of Gram-positive (BS and BA) and Gram-negative (EA and EC) bacteria. The bioactivity of these phytoextracts makes them potential candidates for various applications, such as additives to food, enhancing polyphenol content, and serving as an antioxidant. Comparatively, the overall inhibition of various bacteria by the extract against the selected microbes was lower for HT than for CSS and LP (Figure 1). 

The inhibition of the three extracts varied according to their source and concentration. The CSS extract demonstrated high potency against EA, EC, BS, and BA at 100 and 50 mg/mL in water and methanol, respectively. EC growth was more compromised by the CSS extract compared to its effects on EA, BA, and BS at both 100 and 50 mg/mL in methanol and water. Except for ampicillin (positive control), which exhibited a higher inhibition rate than CSS, 70% methanol and water (negative control) showed poor or no inhibition rate (Figure 1A).

Moreover, the inhibition of HT varied according to concentrations of 100 mg/mL and 50 mg/mL in water and methanol, respectively. While 100 mg/mL HT highly inhibited the growth of EC, EA, BA, and BS, 50 mg/mL HT was more effective against EC compared to other concentrations (Figure 1B). The marginal inhibition rate of the control group was similar to that observed in Figure 1A.

Similarly, the rate of inhibition of LP against the four model organisms showed effective inhibition against BA at 100 mg/mL in 70% methanol, compared to the rest of the concentrations in all the organisms, but was less active than the positive antibiotic control. It was also observed that 50 mg/mL of LP inhibition by EA and BA was almost the same but at a lower rate than the positive control and significantly higher than the controls with 70% methanol and water only, respectively (Figure 1C). These results are consistent with those of previous studies on coffee extracts [24,25].

Prior studies had not compared the activities of industrial food production wastes under the same conditions for their possible use economically and sustainably such as considering different concentrations and solvents as was carried out in this study. However, a study found that the antibacterial activities of hot trub with different fractions showed some level of bioactivity against both, Gram-positive and Gram-negative bacteria [16]. This analysis does not compare the most economical raw materials for industrial application. Therefore, a deeper analysis of various food production wastes under the same conditions is necessary for their realistic industrial application. 

Both Gram-positive and Gram-negative pathogens were considered in the current study for determining the MICs of CSS, HT, and LP extracts. The tested pathogens were *Listeria monocytogenes*, *Vibrio parahaemolyticus*, *Cronobacter sakazakii*, *Pseudomonas aeruginosa*, *Salmonella enterica*, *Klebsiella pneumonia*, and *Staphylococcus aureus*. Following the quantitative serial dilution assay in a 96-well microtiter plate, the MIC values of each extract was determined (Table 1).

Antibacterial activity against the two Gram-positive bacteria (*Listeria monocytogenes* and *Staphylococcus aureus*) were assessed and MIC values were determined (Table 1). The 70% MeOH-dissolved CSS extract had the lowest MIC (3.1 mg/mL), making it the strongest inhibitor of growth for the two pathogens with slightly less efficiency than the water-dissolved CSS extract. The latter showed MIC values against *Listeria monocytogenes* and *Staphylococcus aureus*, with values of 6.3 mg/mL and 12.5 mg/mL, respectively (Table 1). This suggested that water-dissolved CSS extract contained different or lower concentrated antibacterial compounds suggesting a differential extract efficiency for both solvents, which is not surprising. Considering the susceptibility of Gram-negative pathogens to CSS extracts, *Vibrio parahaemolyticus* had the lowest MIC (Table 1). 

Similarly, the LP extract showed the highest potency against *Listeria monocytogenes* with MICs of 3.2 mg/mL and 6.3 mg/mL for the 70% MeOH- and water-dissolved extracts, respectively. Except for *Chronobacter sakazakii* and *Salmonella enterica* with the same MIC susceptibility for LP between 70% MeOH and water extracts, the rest of the MICs for all the pathogens increased two-fold compared to 70% MeOH and water, respectively. This observation could be expected since the composition of the compounds in the dissolved extracts with 70% methanol might differ from that in water and might actually be higher in amount or diversity in each solvent.

In addition, the best MIC exhibited by HT was at 0.4 mg/mL for 70% MeOH dissolved extracts and water-dissolved extracts against *Listeria monocytogenes* and *Pseudomonas aeruginosa*, respectively (Table 1). These two bacteria recorded the highest susceptibility towards HT extract in both water and 70% MeOH. *Vibrio parahaemolyticus* had MICs of 1.6 mg/mL, 6.3 mg/mL, and 1.6 mg/mL for the 70% MeOH-dissolved phytoextract in CSS, LP and HT, respectively. The phytoextracts dissolved in water performed with 6.3 mg/mL, 12.5 mg/mL, and 6.3 mg/mL for CSS, LP, and HT, respectively. The performance of *Vibrio parahaemolyticus* gives the crude extract more relevance for serving as antibacterial activity towards hygiene product formulation as a means of waste valorization principle. *Vibrio parahaemolyticus*, though a Gram-negative bacterium, performed better than the inhibition performance of *Listeria monocytogenes* in terms of MICs (Table 1). Similarly, *Cronobacter sakazakii* performed weakly when compared with the other pathogens considering the 6.3 mg/mL MIC for CSS and HT phytoextract dissolved in 70% MeOH with the rest being 12.5 mg/mL for the phytoextract (Table 1). Similarly, there was a comparable performance of *Cronobacter sakazakii* MICs to *Kblebsiella pneumoniae* MICs, except CSS phytoextract with a 3.1 mg/mL MIC and a 6.5 mg/mL MIC for the LP phytoextract, the rest having 12.50 mg/mL (Table 1). Both 70% MeOH and water for the CSS, LP, and HT phytoextracts showed the lowest MICs against *Salmonella enterica* with 12.50 mg/mL (Table 1).

In addition, their antibacterial activity as crude materials gives them the advantage of being used as active agents for natural products in cosmetics [8]. The above-reported MICs imply possible synergistic effects of various potentially bioactive compounds in the tested extracts contributing to the bioactivity that has been previously reported [26]. The latter study focused on plant-borne phenolics and their antibacterial activities. The results confirmed previous studies, which analyzed the bioactivity of similar food production wastes in terms of MIC determination. In this study, the achieved MIC values were lower than those reported for the total crude extracts. MICs of 37.5 mg/mL and 75 mg/mL were reported for *Staphylococcus aureus* and *Pseudomonas aeruginosa*, respectively, from coffee aqueous extracts and, in another study, showing MICs > 200 and suggesting the MIC value for coffee silverskin to be above 512 mg/mL [10]. Additionally, the organisms considered in this study were mainly food pathogens, including *Listeria monocytogenes*, *Vibrio parahaemolyticus*, *Salmonella enterica*, and *Cronobacter sakazakii*.

Even though the mechanisms of plant-based extracts are not well studied, several pieces of evidence point to the mechanisms of polyphenols as antibacterial agents against microorganisms. Flavonoids, such as quercetin, are known to inhibit DNA gyrase, catechins inhibit the cytoplasmic membrane function of microorganisms, and chalcones (such as xanthohumol) inhibit the energy metabolism of microorganisms, leading to cell death [27,28,29]. The numerous bioactive compounds characterized in each phytoextract (CSS, HT, LP) point to the efficient antibacterial activities observed in each extract against the seven pathogenic organisms. However, the antimicrobial effect cannot be attributed to a single compound in the extract but a concomitant effect with multiple mechanisms participating in the killing of the bacteria. Polyphenols inhibit microbial enzymes by reacting with sulfhydryl groups or through nonspecific protein interactions [30]. Another mechanism of polyphenol compounds against microbes is to react with protein sulfhydryl groups, generating phenolic toxicity and hindering microbial growth [31].

### 3.2. Antioxidant Sum Parameters of Extracts

The extraction yields of the food production waste were successful, with high amounts obtained from each extract: CSS (20 mg/1 g), HT (331 mg/1 g), and LP (482 mg/1 g) (Figure 2A). Additionally, the extraction yields indicated significant differences between the means, with *p* < 0.05.

Polyphenolic compounds are vital plants secondary metabolites which have important functional properties such as antioxidants or antibacterial properties. Spectrophotometric determination using the Folin–Ciocalteau, DPPH, and ABTS assays constitute useful parameters for estimating total polyphenol content (TPC) and antioxidant capacities. Using gallic acid equivalent (GAE) as a proxy, the assay was employed on the three bioactive phytoextracts. 

The extraction yields of the food production waste were successful, with high amounts obtained from each extract: CSS (20 mg/1 g), HT (331 mg/1 g), and LP (482 mg/1 g) (Figure 2A). Additionally, the extraction yields indicated significant differences between the means, with *p* < 0.05. 

The TPC content obtained included 140 mg GAE g^−1^, 100 mg GAE g^−1^, and 20 mg of GAE g^−1^ CSS, LP, and HT, respectively (Figure 2B). Among the three samples, the TPC was higher in CSS if compared to LP. HT showed the lowest TPC values the least. The different food production waste was independent of the TPC with *p* ˂ 0.05.

The ABTS antioxidant activity of the extracts of each waste material was defined by the IC_50_ value. The amount of each extract activity showed that CSS recorded the highest radical scavenging ability of 1.88 mg/mL as against LP with 3.08 mg/mL, and HT with the lowest of 12.93 mg/mL IC_50_, respectively. However, comparing the statistical means revealed that the extract’s ABTS activities were significantly different from each other with *p* ˂ 0.05. 

The results revealed a higher difference among the phytoextracts, such as HT compared to CSS and LP. As shown in Figure 2C,D, CSS had the highest radical scavenging of 9.94 g-GAE/kg followed by LP with 9.77 g-GAE/kg and HT with the lowest level at 1.45 g-GAE/kg (Figure 2D). In addition, the mean of the three industrial wastes shows them to be significantly different from each other with *p* ˂ 0.05. 

Figure 2B–D, respectively, illustrates TPC, ABTS, and DPPH scavenging activity. The higher the total phenolic content in the extract, the higher its corresponding radical scavenging activity in both ABTS and DPPH, suggesting that the most active class of compounds responsible for the extracts could be phenolic compounds. ABTS and DPPH are among the most used in vitro assays for antioxidant sum parameters, based on the neutralization of radicals by antioxidant compounds. ABTS works as an antioxidant via a reaction with an organic radical. ABTS is a good method for the measurement of bioactive compounds such as polyphenols and flavonoids [18,32]. DPPH has an intense deep purple color. In assays, the DPPH radical is neutralized by accepting either a hydrogen atom or an electron from an antioxidant species, converting it into a reduced form (DPPH or DPPH-H). The TPC obtained in the three-food production wastes correlates with the antioxidant activity of ABTS and DPPH (Figure 2B–D). It was also observed that the higher the polyphenolic content, the higher the antioxidant activity. The ABTS cationic radical generated is soluble in both organic and aqueous media, unlike DPPH, which dissolves only in organic media. Consequently, ABTS can screen both hydrophilic and lipophilic compounds, unlike DPPH, as observed in the results, confirming the compounds participating in the bioactivity of the extract are polyphenols from the extracts. 

### 3.3. In Vitro α-Amylase Inhibition Activity

A common property of plant polyphenols is their ability to non-selectively bind to proteins through H-bonds and π-interactions. This phenomenon often leads to the loss of enzyme functions. As a model, the enzyme α-amylase was chosen to further characterize the extracts under investigation, serving as a powerful proxy for the protein-binding properties of the extracts. The polyphenols with antibacterial and antioxidant potential exhibited by the food production waste extract demonstrated relevance for their enzyme inhibition.

The percentage of α-amylase inhibitory activity was analyzed for the three extracts (Figure 3) using α-amylase. Comparing the three industrial waste extracts, they exhibited difference significance with *p* ≤ 0.05. Using 50 mg/mL of the LP extract exhibited the highest inhibition rate of 92.14%, while CSS and HT displayed almost the same inhibition effect with mean percentages of 60.12% and 61.01%, respectively. 

Controlling the release of glucose from dietary carbohydrates is one way to reduce blood glucose levels in patients with type II diabetes mellitus. This may be achieved by inhibiting carbohydrate hydrolyzing enzymes in the digestive organs such as α-amylase, which catalyzes the hydrolysis of starch into maltose and glucose [33]. Synthetic α-amylase inhibitors such as acarbose control postprandial hyperglycemia [34].

### 3.4. UHPLC-ESI-QTOF-MS Characterization of Compounds in Phytoextracts of the Three Food Production Wastes

The database search revealed 30 compounds identified in CSS, 22 in HT, and 34 in LP. Figure 4, Figure 5 and Figure 6 shows structures of some selected identified compounds from the extracts Figure 7 shows a representative base of peak chromatograms from the extracts Compounds with errors above 5 ppm were not considered.

#### 3.4.1. Characterization of Coffee Silverskin Phytoextract

The 30 compounds were identified in CSS following database matching (Figure 4). Some of the main compounds are chlorogenic acid eluted at 20.5 min with *m*/*z* of 353.0851 (C_16_H_18_O_7_). This peak was identified as chlorogenic acid (5-caffeoylquinic acid) owing to the matching fragment ions at the base peak at 191.0542, the other peaks include 85.0289, 87.0071, and 93.0327. Similarly, 1, 3, 7-trimethyluric acid was tentatively assigned to the parent ion at 209.0667 (C_7_H_13_O_7_) with its base peak fragment ion occurring at 137.0222 with other peaks at 79.9583 and 179.0168 occurring at 13.6 min. Hyperoxide appeared at the beginning of the chromatogram at 4.4 min with a main peak at 463.0882 (C_21_H_19_O_12_) and with a base peak at 300.0258 and other fragment ions at 271.0236 and 301.0312. Table 2 provides detailed information on all the compounds identified in the CSS extract [35]. Chlorogenic acids predominantly identified in coffee silverskin have been shown in previous work to possess antiviral activity [36].

The derivatives of chlorogenic acids such as caffeoylquinic acids: 3-caffeolyquinic acid, 5-caffeolyquinic acid, 3,5-dicaffeolyquinic acid, 3,4-dicaffeoylquinic acid, 3,5-di caffeoylquinic acid, and 4,5-di-caffeoylquinic acid are known bioactive compounds locked in CSS and require recovery and potential application [10,37]. The presence of these compounds in CSS confirms the possible synergistic phytoextract bioactivity against microorganisms. It also implies the extraction method used sustained the bioactive compounds without losing their biological properties. CSS is a byproduct of coffee production, which has been suggested as an antibacterial agent [10]. 

#### 3.4.2. Characterization of Hot Trub Phytoextract 

In hot trubs, the most known compounds are usually the remaining bitter acids from the hops. The chromatogram confirmed the presence of most of these compounds in the data analysis. A typical example is Xanthohumol with an *m/z* peak at 353.1384 (C_21_H_21_O_5_) and fragment ions at 119.0498 (base peak), 175.0032, and 163.0025, which occur at 50.7 min. Cohulupone with an *m*/*z* peak at 317.1758 (C_19_H_25_O_4_) occurred towards the end of the chromatogram at 49.9 min and characteristic fragment ions at 205.0863 (base peak), 203.0449, and 133.0652. Similarly, the prenylated compound 8-Prenylnarigenin was identified with an *m/z* peak at 339.1238 (C_20_H_19_O_5_) and its characteristic peaks at 119.0496 (base peak), 133.0652, 219.0630, and 93.0333. Most of these compounds have already been identified in hot trub [16]. Hop compounds have been investigated for their antibacterial activities. However, there is limited research on their broad spectrum of antibacterial and antifungal activities, as well as the yield of these extracts compared to other waste.

Table 3 indicates the detailed information of each fragmentation pattern peak. The total number of compounds identified and characterized in HT was 22 (Table 3). The identified compounds were predominantly α and ß—acids of hop, which was expected due to the partial insolubility of bitter acids during beer production [16]. It has been established that the alpha acid present in beer production usually absorbs yeast cells, trub, and other materials causing it to undergo different chemical changes to form several compounds [38]. The extraction procedure requires a controlled manner to recover the majority of these compounds in HT. Of the 22 compounds identified in HT, most of them have been reported to be bioactive and nontoxic to several bacteria and fungi. HT extracts have shown nontoxicity against various cell lines making them valuable resources for the circular economy[16]. Structures of some of the identified compounds are presented in Figure 5.

#### 3.4.3. Characterization of Lemon Peel Phytoextract

Herein, 34 compounds were identified using LC-ESI-QTOF-MS (Table 4) with literature and database matching. Characterization of the compounds indicated that Vicenin-2, which was previously identified in LP, eluted at 7 min with a major *m/z* mass of 593.1512 (C_27_H_29_O_15_) and its fragment ions at 353.0637 (base peak), 383.0748, and 325.0689. In addition, Lacitrin-O-rutinoside was identified in LP with a main *m/z* of 639.1569 (C_28_H_31_O_17_) and 315.0127 (base peak fragment), and other fragment ions at 330.0362, 316.0184, and 331.0452 at 31.3 min of the chromatogram. Eriodictyol was eluted at 44.2 min with an *m/z* peak at 287.0561 (C_15_H_11_O_6_) and its characteristic fragment ions were 135.0435 (base peak), 134.0365, 83.0117, and 151.0054 [39]. The structures of some of the compounds identified in the LP extract are shown in Figure 6.

Figure 7 indicates the base peak chromatogram of the three food waste valorizations considered in this study. The chromatographic diagram shows several peaks of high intensity and has been characterized to identify the compounds in these phytoextracts. Details of the compounds are listed in Table 2, Table 3 and Table 4. The structures of selected identified compounds are indicated in Figure 4, Figure 5 and Figure 6. The presence of these compounds in the phytoextracts demonstrates the retention of several bioactive compounds after the extraction process. The chromatogram represents the intensity and retention time of the identified compounds in each extract including the coffee silverskin base peak chromatogram (Figure 7A), hot trub phytoextract base peak chromatogram (Figure 7B), and lemon peel base peak chromatogram (Figure 7C).

### 3.5. Cell LDH Cytotoxicity Activity of Phytoextracts

The safety of a product for target users is crucial for preventing negative side effects. The first point of contact of the product is the skin, which needs assessment of its safety; hence the need for this cytotoxicity testing. HaCaT cells are a model cell line regarded as suitable for in vitro testing of the cytotoxicity of substances or products intended for dermatological applications. The LDH assay was used to measure cell damage and the release of oxidative stress to increase the cell membrane permeability. As represented in the Appendix A, none of the three extracts showed nontoxicity when compared to the control. In this study, cell cytotoxicity assessment was conducted using an LDH assay against the keratinocyte cell line and HaCaT cells, due to the bioactivity of all the extracts (CSS, HT and LP, Table 1). The positive control used had a higher absorbance at 0.78 nm (results shown in Appendix A). The cytotoxicity assays of the HaCaT cells at varying concentrations of the extracts indicated that none of the extracts’ tested concentrations exceeded the positive control when assessing their toxicity towards HaCaT cells. The varying concentrations of the cytotoxicity testing were considered due to differences in MIC values at different concentrations of extracts and solvents (Table 1). Since the food production extracts are nontoxic to the skin cells (HaCaT cells), we suggest their possible application as disinfectants or for hygiene applications in the food industry. Furthermore, their nontoxicity shows their possible application as food ingredients and ultimately for the isolation of bioactive compounds for pharmaceutical application. It was previously established that extracts from food production waste are nontoxic due to their origin from food sources, such as hot trub [16].

## 4. Conclusions

In conclusion, this research demonstrates that extracts of food production waste materials such as coffee silverskin, lemon peel, and hot trub possess antibacterial activities against both Gram-positive and Gram-negative bacteria. A basic LC-MS-based chemical characterization reveals chlorogenic acids in CSS, flavonoids in LP, and terpenes in HT as contributors to antibacterial activities. Such activities could also be demonstrated against relevant pathogenic bacteria such as *Listeria monocytogenes*, *Staphylococcus aureus*, *Vibrio parahaemolyticus* or *Salmonella enterica*. Experimentally determined MIC values suggest that such extracts could be employed in multiple applications including hygiene applications in the food industry or public spaces such as hospitals or public transport, cosmetic applications in deodorants, or applications in crop protection.

## Figures and Tables

**Figure 1 foods-13-01902-f001:**
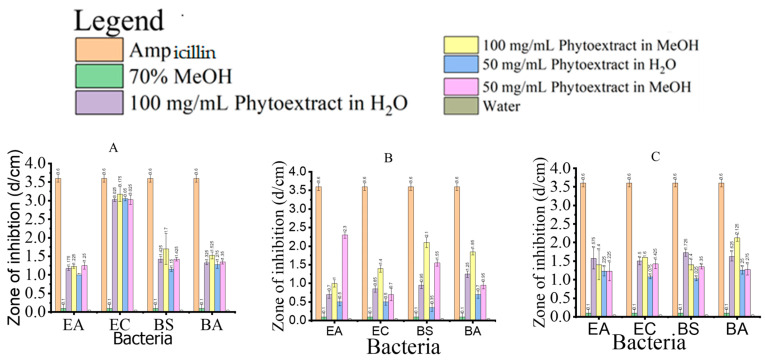
Antibacterial activities of phytoextracts from food production waste against microorganisms (**A**) coffee silverskin extract, (**B**) hot trub extract, (**C**) Lemon peel extract. Error bars represent mean and standard deviation. Bar graphs were drawn using origin software. EA = *Erwinia Amylovora* (Gram −), EC = *E. coli* (Gram −), BS = B. subtilis (Gram +) and *B. aquimaris* (Gram +), and d = diameter.

**Figure 2 foods-13-01902-f002:**
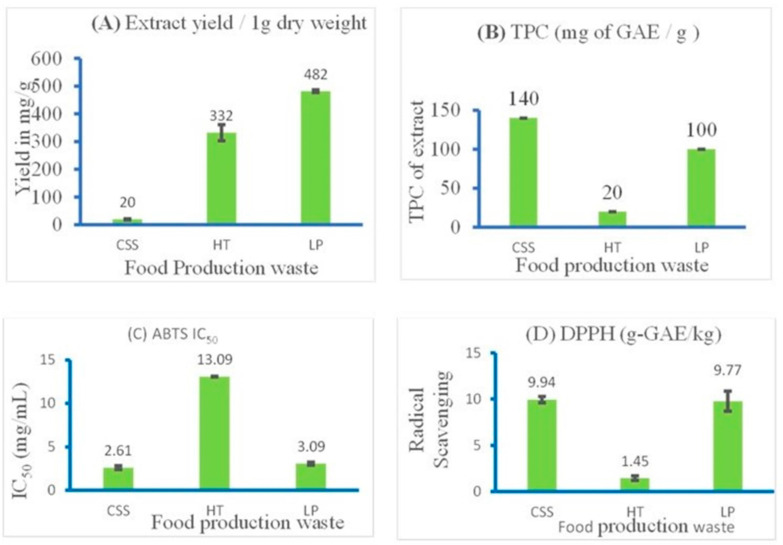
Antioxidant sum parameters of three food production waste extracts: (**A**) yield of crude extract expressed in mg per 1 g of the dry weight of waste, (**B**) total phenolic content (TPC) of different industrial waste extracts expressed as mg of gallic acid equivalent per mg of dry weight, (**C**) (2,2′-azino-bis(3-ethylbenzothiazoline-6-sulfonic acid)) scavenging activities, and (**D**) 2,2-diphenyl-1-picrylhydrazy radical scavenging activities.

**Figure 3 foods-13-01902-f003:**
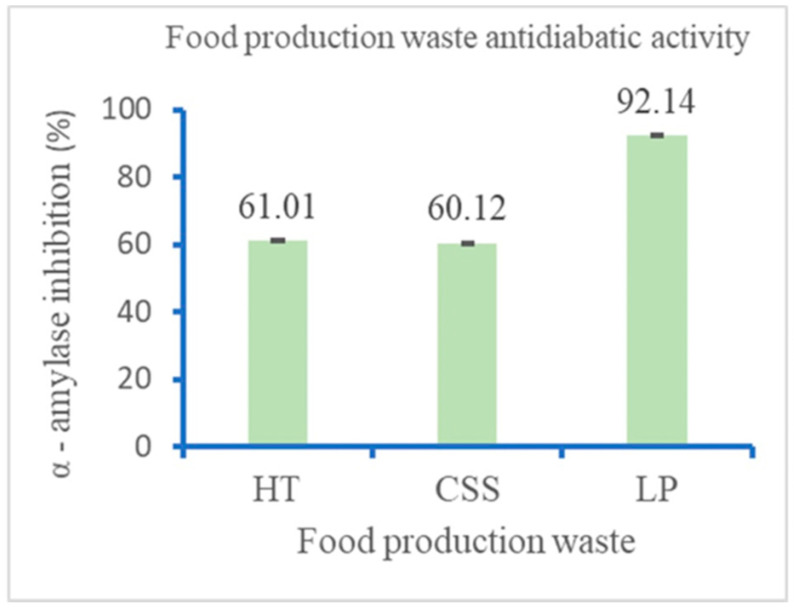
In vitro α-amylase inhibition activity expressed as percentage inhibition: HT (hot trub); CSS (Coffee silverskin); and LP (Lemon peel).

**Figure 4 foods-13-01902-f004:**
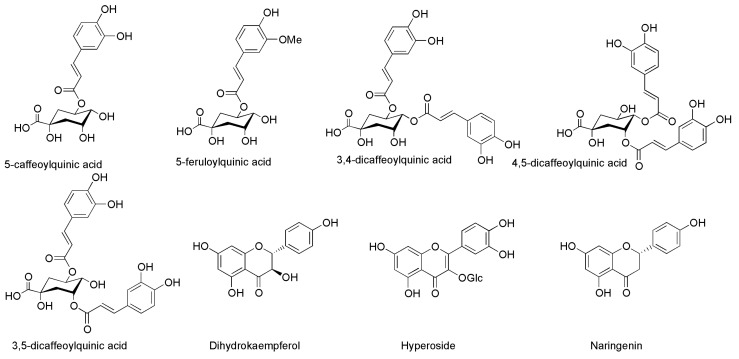
Compounds identified in coffee silverskin phytoextracts by LC-ESI-QTOF-MS. Detailed information can be found in Table 2.

**Figure 5 foods-13-01902-f005:**
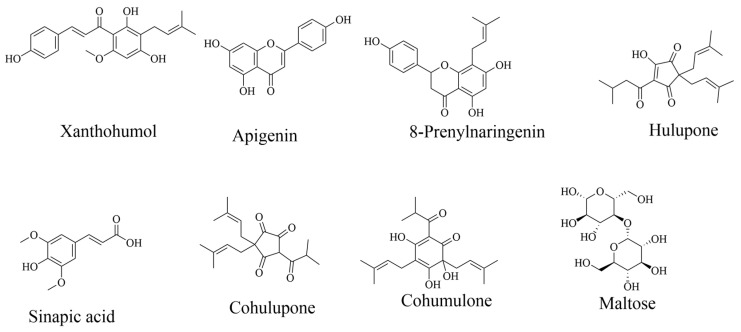
Compounds identified in hot trub phytoextracts by LC-ESI-QTOF-MS. Detailed information in Table 3.

**Figure 6 foods-13-01902-f006:**
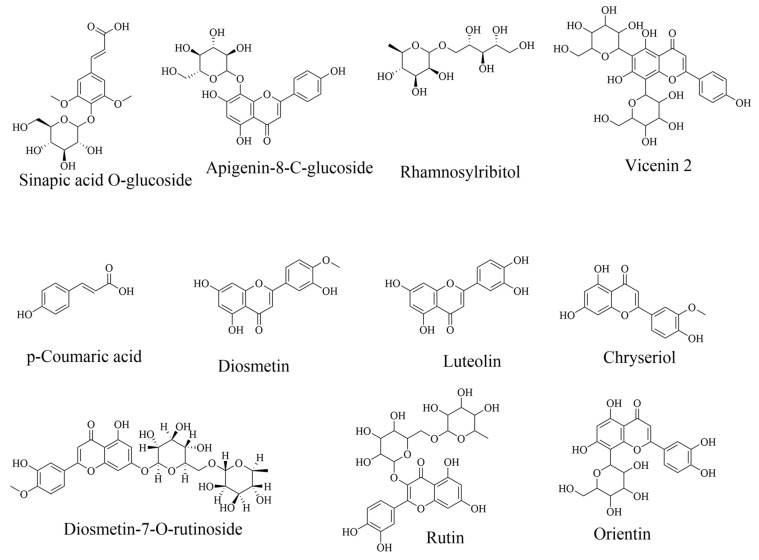
Compounds identified in Lemon peel phytoextracts.by LC-ESI-QTOF-MS. Detailed information in Table 4.

**Figure 7 foods-13-01902-f007:**
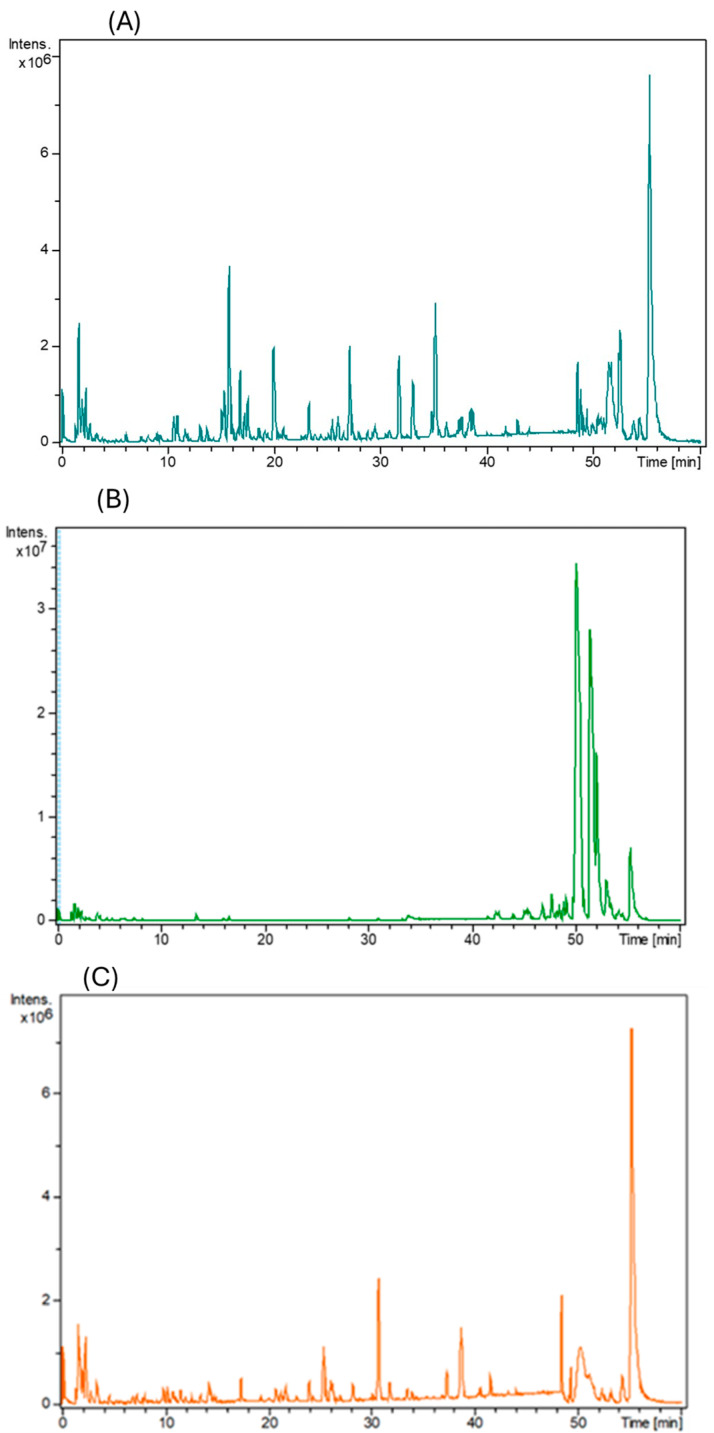
LC-ESI-QTOF-MS base peak chromatograms of phytoextracts in negative ion mode: (**A**) coffee silverskin with details in Table 2, (**B**) hot trub with details in Table 3, and (**C**) lemon peel. All plots represent intensity versus retention time. Details of identified compounds are in Table 2, Table 3 and Table 4.

**Table 1 foods-13-01902-t001:** Minimum inhibitory concentration (MIC) of food production waste phytoextracts against seven pathogenic microorganisms.

	MIC (mg/mL)					
Bacteria	Coffee Silverskin		Lemon Peel		Hot Trub	
	MeOH	H_2_O	MeOH	H_2_O	MeOH	H_2_O
*Listeria monocytogenes +*	3.1	6.3	3.1	6.3	0.4	6.3
*Vibrio parahaemolyticus −*	1.6	6.3	6.3	12.5	1.6	6.3
*Cronobacter sakazakii −*	6.3	12.5	12.5	12.5	6.3	12.5
*Pseudomonas aeruginosa −*	3.1	12.5	6.3	12.5	0.4	0.8
*Salmonella enterica −*	12.5	12.5	12.5	12.5	12.5	12.5
*Klebsiella pneumoniae −*	3.1	12.5	6.5	12.5	12.5	12.5
*Staphylococcus aureus +*	3.1	12.5	6.3	12.5	3.1	6.3

Minimum inhibitory concentration (MIC) of three food production waste crude extracts dissolved in 70% methanol (MeOH) and water (H_2_O): + = Gram-positive pathogenic organism and *−* = Gram-negative organism.

**Table 2 foods-13-01902-t002:** Database identification of compounds from coffee silverskin methanolic extract using LC-ESI-QTOF-MS with their MS fragments in negative ion mode.

#	RT/min	[M − H]	Err/ppm	Expemental *m/z*	Theo. *m/z*	Compound	Base Peak Ion		Other Fragments Ions		
1	36.3	C_21_H_19_O_11_	4.0	447.0915	447.0933	Kaempferol-3-O-glucoside	227.0333	255.0283	284.0310	256.0325	96.9590
2	32.5	C_21_H_19_O_12_	4.4	463.0862	463.0882	Hyperoside	300.0258	271.0236	255.0292	301.0312	272.0269
3	34.7	C_25_H_23_O_12_	5.4	515.1167	515.1195	3,4-diCQA	173.0446	191.0547	135.0452	179.0330	161.0226
4	35.5	C_25_H_23_O_12_	3.6	515.1177	515.1195	3,5-diCQA	191.0549	135.0436	179.0344	136.0472	192.0574
5	37.9	C_25_H_23_O_12_	5.4	515.1167	515.1195	4,5-diCQA	173.0443	179.0344	135.0439	191.0552	93.0336
6	29.7	C_34_H_25_O_11_	−2.4	609.1417	609.1402	Kaempferol deivative	284.0306	255.0270	285.0390	227.0333	319.1952
7	31.4	C_34_H_25_O_11_	−4.1	609.1428	609.1402	Rutin	300.0255	301.0310	271.0242	255.0284	302.0355
8	15.8	C_16_H_17_O_9_	4.7	353.0851	353.0878	5-cafeoyl quinic acid	191.0542	85.0289	87.0071	93.0327	192.0589
9	17.5	C_16_H_17_O_9_	4.3	353.0856	353.0878	3-cafeoyl quinic acid	135.0434	93.0331	191.0543	173.0422	136.0497
10	36.3	C_21_H_19_O_11_	3.3	447.0905	447.0933	Kaempferol 3-O-glucoside	227.0336	255.0282	284.0314	256.0322	285.037
11	23.2	C_17_H_19_O_9_	4.5	367.1003	367.1003	5-O-Feruloylquinic acid	191.0557	134.0353	93.0334	87.0088	173.0438
12	27.2	C_17_H_19_O_9_	4.9	367.1003	367.0976	4-O-Feruloylquinic acid	191.0538	85.0288	93.0339	192.0565	87.0089
13	21.9	C_17_H_19_O_9_	4.7	367.1035	367.1006	3-O-Feruloylquinic acid	93.0334	134.0355	173.0444	111.0441	94.0363
14	15.8	C_39_H_31_O_13_	0.3	707.1768	707.1770	4-cafeoyl quinic acid	191.0547	192.0581	161.0229	179.0336	93.0343
15	19.9	C_12_H_15_O_7_	5.0	271.0799	271.0823	Arbutine	137.0250	149.0203	152.0466	123.0077	138.0293
16	47.5	C_15_H_11_O_5_	3.8	271.0602	271.0612	Naringenin	119.0490	93.0336	83.0139	107.0113	161.0650
17	34.5	C_15_H_17_O_8_	5.2	325.0912	325.0929	p-Coumaric acid-O-hexoside	78.9593	292.8046	229.8419	308.8069	102.9474
18	2	C_13_H_21_O_12_	5.1	369.102	369.1038	Fraxetin-8-O-glucoside	191.0563	85.0290	103.0030	87.0090	129.0189
19	1.9	C_13_H_21_O_12_	4.0	369.1024	369.1038	7-hydroxy-6-methoxy-8-[3,4,5-trihydroxy-6-(hydroxymethyl)(2H-3,4,5,6-tetrahydr opyran-2-yl)oxy]chromen-2-one	191.0557	85.0287	87.0083	192.0601	83.0495
20	23.2	C_17_H_19_O_9_	4.7	367.1003	367.1035	Methyl 5-O-caffeoylquinate	191.0556	134.0353	93.0334	87.0088	173.0438
21	18.5	C_9_H_7_O_4_	3.8	179.0334	179.0350	Cafffeic acid	134.0362	135.0424	89.0394	106.0431	118.0379
22	25.1	C_16_H_17_O_8_	5.0	337.0899	337.0829	Coumaroyl quinic acid (isomer)	191.0543	85.0286	93.0354	111.0440	618.7287
23	27.2	C_17_H_19_O_9_	2.8	367.1009	367.1035	5-O-Feruloylquinic acid	191.0560	85.0293	93.0338	134.0335	127.0385
24	301	C_16_H_13_O_6_	5.0	301.07	301.0718	Quercetin	151.0051	164.0108	108.0195	136.0195	135.0400
25	37.1	C_15_H_11_O_6_	4.4	287.0537	287.0561	Dihydrokaempferol	125.0230	151.0065	133.0281	152.0089	149.9032
26	21.8	C_16_H_17_O_8_	4.8	337.0899	337.0929	4-O-p-Coumaroylquinic acid	93.0335	173.0462	119.0486	292.8072	111.0442
27	20.5	C_16_H_17_O_9_	3.8	353.0851	353.0878	Chlorogenic acid	191.0542	85.0289	87.0071	93.0327	192.0589
28	20.6	C_16_H_17_O_8_	5.0	337.0909	337.0929	(1R,3R,4S,5R)-1,3,4-trihydroxy-5-[(E)-3-(4-hydroxyphenyl)prop-2-enoyl]oxycyclohexane-1-carboxylic acid	93.0341	191.0544	119.0483	292.8119	127.0381
29	13.6	C_7_H_13_O_7_	3.2	209.066	209.0667	1,3,7-trimethyluric acid	137.0222	78.9583	179.0168	124.0505	80.0015
30	27.1	C_17_H_19_O_9_	4.4	367.1004	367.1034	5-O-Feruloylquinic acid	191.0537	85.0288	93.0339	192.0565	87.0089

RT = retention time, Err = error, ppm = parts per million, Theo = Theoretical, 3,5-diCQA = 3,5-Dicaffeoylquinic acid: 3,4-diCQA = 3,4-Dicaffeoylquinic acid: 4,5-diCQA = 4,5-Dicaffeoylquinic acid.

**Table 3 foods-13-01902-t003:** Comprehensive identification of Hot Trub compounds using LC-ESI-QTOF-MS with their MS fragments in negative ion mode.

#	RT/min	[M − H]	Err/ppm	Exp. *m/z*	Theo. *m/z*	Compound	Base Peak Ion		Other Fragments Ions			
1	52	C_25_H_27_O_5_	2.7	407.1853	407.1865	8-Geranylnaringenin	119.0496	287.1283	133.0649	201.0543	93.0338	201.0183
2	48.7	C_18_H_33_O_5_	5.3	329.2316	329.2347	Desdimethyl-octahydro-iso-cohumulone	139.1130	211.1329	171.0998	127.1125	172.1068	99.0780
3	50.4	C_20_H_19_O_5_	4.6	339.1222	339.1238	Flavaprenin/8-Prenylnaringenin	119.0498	133.0652	219.0630	93.0333	176.0108	151.0758
4	49.1	C_21_H_21_O_5_	5.1	353.1373	353.1394	Isoxanthohumol	119.0496	120.0529	133.0651	163.0025	175.0031	165.0904
5	50.7	C_21_H_21_O_5_	3.5	353.1382	353.1384	Xanthohumol	119.0498	175.0032	163.0025	120.0526	203.0335	190.0636
6	27.3	C_26_H_27_O_14_	3.8	563.1385	563.1406	Apigenin 6-C-pentosyl-8-C-hexoside	353.0654	383.0745	384.0734	365.0665	413.0918	296.0616
7	47.5	C_15_H_9_O_5_	4.3	269.0439	269.055	Apigenin	117.0337	78.9583	149.0241	260.0873	159.0433	180.0546
8	45.4	C_20_H_27_O_5_	3.8	347.184	347.1864	Cohumulone I	125.0595	263.0913	233.1171	261.1107	193.0491	221.0446
9	49.9	C_19_H_25_O_4_	4.2	317.1745	317.1758	Cohulupone	205.0863	133.0652	205.0499	152.0472	111.0446	233.0809
10	51.2	C_20_H_27_O_4_	4.2	331.1901	331.1915	Hulupone/Adhulupone	219.1014	125.0605	219.0651	191.0709	166.0629	247.0963
11	52.9	C_21_H_29_O_5_	3.3	361.2008	361.2020	n-Cis-isohumulone	195.0657	125.0602	223.0602	196.0697	153.0187	163.0754
12	52.8	C_21_H_29_O_5_	3.1	361.2009	361.2020	n-trans-isohumulone (isomer)	195.0658	125.0599	223.0604	196.0704	153.0185	179.0706
13	52.6	C_21_H_29_O_5_	5.0	361.2009	361.2020	trans-isoadhumulone (isomer)	195.0649	125.0610	205.0866	289.1428	167.0686	203.0695
14	52.2	C_21_H_29_O_5_	3.7	361.2007	361.2020	Iso-α-n/ad-humulone (isomer)	219.1011	303.1591	245.0787	125.0602	219.0624	259.1727
15	54.1	C_26_H_37_O_5_	5.0	429.2622	429.2646	Hydroxytricyclolupone/Hydroxytricycloadlupone	125.0599	245.1526	259.0972	176.0836	99.0803	78.9582
16	52.9	C_25_H_35_O_5_(-CH_3_)	3.2	415.2477	415.2490	Hydroxytricyclocolupone epimers	111.0433	245.1485	148.0872	107.0486	125.0990	149.0980
17	26.1	C_27_H_29_O_15_	4.1	593.1476	593.1512	Apigenin-C-hexoside-O-hexoside	311.0528	297.0374	282.0499	237.0897	283.0580	298.0407
18	52.9	C_21_H_29_O_5_	3.6	361.2007	361.2020	cis-isoadhumulone	195.0657	125.0604	223.0601	196.0696	153.0188	163.0756
19	52	C_25_H_35_O_5_	1.8	415.2482	415.2490	4-Hydroxycoluplone/Hydroxytricyclocolupone	111.0446	259.0946	181.0502	149.0968	203.1417	209.0455
20	43.9	C_18_H_23_O_5_	4.1	319.1525	319.1551	Deisopropyltricycloisohumulone	125.0607	319.1534	137.0963	165.0912	78.9597	554.4515
21	28.8	C_11_H_11_O_5_	4.2	223.0594	223.0612	Sinapic acid	93.0331	94.0354	185.4320	121.0275		
22	1.8	C_12_H_21_O_11_	3.9	341.1076	341.1089	Maltose	101.0238	161.8954	78.9558	103.0007		

RT = retention time, Err = error, ppm = parts per million, Theo = Theoretical, Exp. = experimental.

**Table 4 foods-13-01902-t004:** Comprehensive identification of lemon peel compounds using LC-ESI-QTOF-MS with their MS fragments in negative ion mode.

#	RT/min	[M − H]	Err/ppm	Exp. *m*/*z*	Theo. *m*/*z*	Compound Name	Base Peak Ion				Other Fragments Ions			
1	42.9	C_19_H_29_O_8_	3.6	385.1831	385.1868	Sinapic acid O-hexoside	99.0418	107.0480	522.6395	87.0098	179.1412	92.9205	101.0220	302.7875
2	32	C_26_H_25_O_14_	4.4	561.1214	561.1250	courmaric acid ester	223.0615	129.0186	85.0284	147.0313	125.0232	83.0142		
3	50.5	C_18_H_31_O_4_	2.2	311.2208	311.2228	(E)-octadec-8-enedioic acid (fatty acid)	183.0121	96.9601	79.9546	74.9865	155.1113	78.9606		
4	38.6	C_28_H_33_O_15_	4.8	609.179	609.1825	Neohesperidin	301.0693	286.0459	284.0307	302.0731	299.0540	242.0559	257.0795	164.0102
5	31.3	C_27_H_29_O_16_	3.5	609.1421	609.1461	Rutin	300.0253	301.0305	271.0227	255.0288	302.0338	272.0296	151.0023	303.0366
6	40	C_28_H_31_O_15_	3.6	607.1647	607.1668	Diosmetin-7-O-rutinoside	299.0521	284.0289	285.0333	301.0691	300.0568	112.9823	211.0702	1216.0304
7	30.5	C_27_H_31_O_15_	3.8	595.1628	595.1668	Neoeriocitrin	151.0028	135.0436	125.0238	175.0031	107.0128	287.0542	136.0471	101.0214
8	31.2	C_28_H_31_O_17_	2.2	639.1553	639.1567	Laricitrin-O-rutinoside	315.0127	330.0362	316.0184	331.0452	287.0171	271.0228	209.0082	243.0254
9	28.7	C_11_H_11_O_5_	4.4	223.0598	223.0612	Sinapic acid	91.0604	93.0333	94.0362	98.0250	105.0341	121.0273	135.0421	148.0176
10	23.9	C_27_H_29_O_15_	5.0	593.147	593.1512	Vicenin 2	353.0637	383.0748	325.0689	354.0667	384.0769	297.0734	365.0613	473.1064
11	30.9	C_21_H_19_O_10_	4.1	431.0945	431.0984	Apigenin 8-C-glucoside	283.0590	311.0529	284.0635	117.0330	293.0470	282.0494	312.0552	163.0385
12	25.3	C_16_H_21_O_9_	3.8	357.1156	357.1191	3-(2-Glucosyloxy-4-methoxyphenyl)propanoic acid	151.0742	136.0524	121.0279	177.0529	112.9288	158.9271	371.7675	153.0899
13	26.7	C_33_H_23_O_10_	2.8	579.1281	579.1297	4′,7,7″-Trimethoxyamentoflavone	298.0433	309.0382	327.0507	297.0370	459.0974	285.0439	351.0459	299.0575
14	37.1	C_27_H_31_O_14_	5.5	579.1688	579.1719	Naringin	271.0575	151.0035	119.0480	272.0589	177.0147	107.0117	165.0187	295.0477
15	31.2	C_28_H_31_O_17_	2.2	639.1553	639.1567	Laricitrin-O-rutinoside	315.0127	330.0362	316.0184	331.0452	287.0171	271.0228	209.0082	243.0254
16	31.6	C_21_H_21_O_11_	5.1	449.1067	449.1089	Eriodictyol-7-O-glucoside	151.0022	135.0435	107.0128	174.9997	109.0280	136.0471	193.0118	165.0160
17	28.1	C_21_H_19_O_11_	4.2	447.0901	447.0933	Orientin	327.0494	299.0524	297.0381	298.0481	284.0303	78.9576	328.0569	311.0558
18	27.4	C_21_H_19_O_11_	3.6	447.0903	447.0933	Kaempferol-3-O-glucoside	298.0443	297.0377	299.0532	327.0470	285.0390	284.0310	311.0510	357.0572
19	30.1	C_34_H_41_O_20_	3.0	769.2174	769.2197	Isorhamnetin-3-O-2G-rhamnosylrutinoside	314.0408	299.0167	315.0456	300.0260	271.0231	1226.9550	292.8148	243.0260
20	25.2	C_28_H_31_O_16_	4.7	623.157	623.1618	Chrysoeriol 6,8-di-C-glucoside	383.0740	413.0841	312.0619	384.0789	414.0877	503.1185	395.0683	313.0642
21	27.1	C_33_H_39_O_20_	2.4	755.2022	755.2040	Quercetin-3-O-(2G-α-L-rhamnosyl)-rutinoside	300.0255	301.0290	271.0226	137.0222	255.0281	235.1170	299.0172	112.9855
22	48.1	C_16_H_11_O_6_	3.7	299.0535	299.0561	Diosmetin	284.0300	256.0350	285.0381	255.0296	151.0035	257.0373	286.0348	211.0392
23	48	C_16_H_11_O_6_	5.0	299.0534	299.0561	Chrysoeriol	284.0318	256.0350	107.0120	280.8580	255.0334	150.9987	211.0369	227.0392
24	39.2	C_22_H_21_O_11_	4.2	461.1061	461.1089	Peonidin-3-O-beta-galactoside	283.0225	255.0276	284.0270	298.0442	297.0391	256.0311	446.0877	299.0493
25	20.9	C_17_H_21_O_10_	4.9	385.111	385.1140	Sinapoyl-D-glucose	190.0262	175.0025	191.0299	205.0488	164.0448	119.0129	147.0070	95.0128
26	10.29	C_25_H_17_O_5_	-3.7	397.1112	397.1081	Bis[4-(3-hydroxyphenoxy)phenyl]methanone	125.0238	78.9579	158.9249	217.0484	615.6301	300.8143	102.9487	126.0226
27	31.2	C_27_H_29_O_14_	3.7	577.1541	577.1563	isovitexin 2″-O-rhamnoside	293.0431	311.0565	294.0483	323.0534	341.0650	281.0422	269.0397	295.0561
28	35.9	C_27_H_29_O_14_	5.0	577.1534	577.1563	Apigenin-7-O-neohesperidoside	269.0431	270.0440	181.0491	268.0356	387.1107	166.0255	329.0278	205.0855
29	3.7	C_11_H_21_O_9_	2.7	297.1183	297.1191	Rhamnosylribitol	87.0071	147.0285	78.9589	106.0278	86.9379	115.0029	229.8425	96.0076
30	35.2	C_35_H_29_O_11_	-2.7	625.1732	625.1774	Kuwanon L	271.0587	151.0022	272.0623	119.0485	313.0689	273.0648	93.0330	107.0126
31	44.9	C_15_H_9_O_6_	4.6	285.0386	285.0405	Luteolin	133.0279	78.9571	151.0013	107.0126	285.0434	217.0509	195.0453	121.0303
32	44.2	C_15_H_11_O_6_	3.3	287.0537	287.0561	Eriodictyol	135.0435	134.0365	83.0117	151.0054	115.9218	107.0132	106.0438	177.0635
33	31	C_26_H_27_O_14_	4.6	563.138	563.1404	Isovitexin 2″-O-arabinoside	293.0440	294.0496	311.0546	295.0525	282.0542	283.0545	323.0548	312.0609
34	12.2	C_15_H_19_O_9_	3.8	343.1008	343.1035	Homovanillic acid O-hexoside	137.0587	121.0271	292.8014	109.0295	135.0435	294.8085	93.0361	78.9579

RT = retention time, Err = error, ppm = parts per million, Theo = Theoretical and Exp. = experimental.

## Data Availability

The original contributions presented in the study are included in the article/Appendix A, further inquiries can be directed to the corresponding author.

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
