# Peer review of "Antibacterial Activity Potential of Industrial Food Production Waste Extracts against Pathogenic Bacteria: Comparative Analysis and Characterization"

_foods, 2024, doi:10.3390/foods13121902_

Round 1

Reviewer 1 Report

Comments and Suggestions for Authors

The work aims to compare the potential of three food production waste extracts (lemon peel, hot trub and coffee silverskin) as sources of bioactive compounds.

The paper is very interesting and detailed

The manuscript is well written and the purpose of the study is very interesting. The introduction provides a sufficient background, the methods are adequately described and the results are clearly presented and discussed in detail.

I have just a few observations to make:

- Are the bacteria used for agar diffusion assay and MIC assay reference strains or wild strains? Please add this information in the Materials and Methods section.

- Line 126: I think that the comma between “only” and “media” is not necessary.

- Please enlarge Figure 1 and Figure 7 to make them easier to read. In Figure 1, please also specify what d in “d/cm” stands for. Is it diameter? Furthermore, the results should be expressed in mm and the numerical values should be reported in the figure or in a table.

Author Response

.Reviewer 1

Reviewer

Responses

. - Are the bacteria used for agar diffusion assay and MIC assay reference strains or wild strains? Please add this information in the Materials and Methods section.

The bacteria are reference bacteria from constructor university lab

- Line 126: I think that the comma between “only” and “media” is not necessary.

Corrected. Thank you

- Please enlarge Figure 1 and Figure 7 to make them easier to read. In Figure 1, please also specify what d in “d/cm” stands for. Is it diameter? Furthermore, the results should be expressed in mm and the numerical values should be reported in the figure or in a table.

Thank you very much. Figure enlarged

Yes, d stands for diameter.

Reviewer 2 Report

Comments and Suggestions for Authors

1.More recent research on waste utilization needs to be highlightd in the Introduction.

2.Necessary references are needed to support the contents (experimenta lconditions, wavenumber, formula, etc) of α-Amylase assay and α-Amylase enzyme bioassay with food production waste extracts.

3.How about the ultrasound power in Extraction procedure?

4.Activity evaluation suggests that at least two concentration levels are set.

5.After identification with ILMS, the main 1-2 compounds  are suggested to be quantified.

6.Different extraction conditions are suggested to be investigated, so that the different extraction yields and activities of the products can be compared.

Comments on the Quality of English Language

Minor revision is required

Author Response

Reviewer 2

Reviewer 2

Responses

1.More recent research on waste utilization needs to be highlightd in the Introduction.

Recent research documents were cited. Thank you very much

2.Necessary references are needed to support the contents (experimenta lconditions, wavenumber, formula, etc) of α-Amylase assay and α-Amylase enzyme bioassay with food production waste extracts

Relevant references were cited where necessary. Thank you

3.How about the ultrasound power in Extraction procedure?

Ultrasound power was not considered in this experiment. Thank you

4.Activity evaluation suggests that at least two concentration levels are set

Yes.

5.After identification with ILMS, the main 1-2 compounds  are suggested to be quantified.

We did not quantify the compounds in this work only profiling

6.Different extraction conditions are suggested to be investigated, so that the different extraction yields and activities of the products can be comared.

For the purpose of this work only one condition was monitored with different waste.

Reviewer 3 Report

Comments and Suggestions for Authors

I am very grateful to you for the invitation to review the manuscript foods-3032421 by Ziemah and coauthors "Antimicrobial Activities Potential of Industrial Food Production Waste Extracts: Comparative Analysis and Characterization”. This work aims to compare the extraction yield of bioactive compounds from three industrial food production wastes. Using various antioxidant assays, the most active waste extract will be identified, and enzyme inhibition activity will be tested. Antibacterial activity will be ascertained against both pathogenic and non-pathogenic organisms. Finally, the compounds in the extracts will be characterized using ultra-high-performance liquid chromatography-electrospray ionization-quadrupole-time of flight-mass spectrometry (UHPLC-ESI-QTOF-MS). The work is interesting but needs adjustments to increase the quality of the material.

Comments:

- Lines 2-4: Specify in the title that the activity is against pathogenic bacteria. "Antimicrobial" is too generic.

- Abstract: Insert a brief sentence about the introductory differential of the work.

- Abstract: Please indicate a better and more detailed step-by-step description of the work.

- Line 15: Was DPPH activity not verified?

- Lines 21-22: Change the repeated keywords to different words from the title.

- Graphical Abstract: Replace commas with periods in the graphs of the Graphical Abstract; use italics for microorganisms and subscripts for chemical formulas.

- Lines 27-29: Although important, I suggest removing the sentence as it is disconnected from the rest of the text.

- Lines 29-31: Please better highlight the amount of waste produced worldwide.

- Introduction: Characterize the chemical components present in these residues that can be reused.

- Lines 47-50: It is necessary to characterize the residues in terms of the quantities generated and their average chemical composition.

- Lines 50-52: The need for new antimicrobials should be emphasized more strongly.

- Lines 64-70: Please standardize the specification of reagents, including the brand.

- Line 75: Is the company name correct?

- Lines 88-89: Check the sentence. Better: "Briefly, 1 g of food production waste was weighed, and 10 mL of a 70% MeOH solution was added. The mixture was then sonicated in a water bath for 15 minutes."

- Lines 92-93: "to remove residual extraction solvents into a powder?" Review the sentence.

- Line 97: Better “antibacterial.” Please check and standardize throughout the text.

- Standardize units throughout the text (h, hour, min, minutes, etc.).

- Lines 227-231: This is unclear. How many residues were used? Previously, three were mentioned, and here, it states ten. Please explain better.

- Lines 227-234: If not all residues are presented, only include the studied material.

- Review the numbering of items, as the article's structure has results before methodology. This applies to the numbering of tables and figures—number according to their order in the text.

- Lines 268-274: Standardize the font size in the sentence.

- Line 276: Change “Figure 1. Figure 1.” to “Figure 1.”

- Figure 1: Enlarge the figure and improve the resolution since the caption is unclear.

- Lines 324-335: Standardize the font size in the sentence.

- Table 1: Change “Hot trob” to “Hot trub.”

- Line 341: This term is unclear. What does “Antioxidant sum parameters of extracts” refer to?

- Figure 2: Improve the resolution of the figure.

- Figure 3: Adjust and standardize the size of the figure.

- Lines 396-403: This material and methods should be removed from the results section.

- Lines 434-435: This sentence needs to be reviewed and elaborated on, as it is disconnected now.

- Line 448: Figure 7B? The sequence of figure numbering is incorrect. The numbering or the way the results are presented must be reviewed.

- Figure 7: Improve the resolution of the figure.

- Results: The authors should discuss the mechanism of bacterial inhibition.

- Results: The discussion on the antioxidant mechanism and the role of the compounds in this process should be expanded.

- Antioxidant activity: Specify the reason for the difference concerning the methods' principles and each antioxidant's characteristics.

- Figure S1: Include supplementary material in a separate file.

Author Response

Reviewer 3 comments and Response

Reviewer 3

Responses

Lines 2-4: Specify in the title that the activity is against pathogenic bacteria. "Antimicrobial" is too generic.

Thank you very much for the suggestion.

The comments have been addressed: lines 1-4

- Abstract: Insert a brief sentence about the introductory differential of the work

Thank you very much suggestion addressed: Line 9-13

- Abstract: Please indicate a better and more detailed step-by-step description of the work.

More details have been added to the abstract. Thank you very much

- Line 15: Was DPPH activity not verified?

DPPH was verified and the results have been added to the abstract as well

- Lines 21-22: Change the repeated keywords to different words from the title.

Line 29-30: Repeated keywords from the title have been replaced. Thank you

- Graphical Abstract: Replace commas with periods in the graphs of the Graphical Abstract; use italics for microorganisms and subscripts for chemical formulas.

Thank you very much for your suggestion all corrections have been made in the graphical abstract, including commas, italics, etc

- Lines 27-29: Although important, I suggest removing the sentence as it is disconnected from the rest of the text.

The suggestion was well taken and the sentence was taken out. Thank you

- Lines 29-31: Please better highlight the amount of waste produced worldwide.

The amount of waste generated annually has been included in the introduction. Thank you

- Introduction: Characterize the chemical components present in these residues that can be reused.

Some chemical compounds known in the waste considered were highlighted in the introduction

- Lines 47-50: It is necessary to characterize the residues in terms of the quantities generated and their average chemical composition.

Average quantities of some chemicals in the residues considered in this manuscript were heightened as suggested. Thank you very much.

- Lines 50-52: The need for new antimicrobials should be emphasized more strongly.

Thank you very much. The need for antimicrobials was emphasized.

- Lines 64-70: Please standardize the specification of reagents, including the brand.

Thank you for pointing out the standardization of the reagents. All reagents used was standardized including brand

- Line 75: Is the company name correct?

The name is correct. Thank you very much

- Lines 88-89: Check the sentence. Better: "Briefly, 1 g of food production waste was weighed, and 10 mL of a 70% MeOH solution was added. The mixture was then sonicated in a water bath for 15 minutes."

Thank you very much: The sentence suggestion was taken.

- Lines 92-93: "to remove residual extraction solvents into a powder?" Review the sentence.

Thank you very much for the correction the sentence was reviewed: Line 121-122

- Line 97: Better “antibacterial.” Please check and standardize throughout the text.

Antibacterial was standardized throughout the manuscript

- Standardize units throughout the text (h, hour, min, minutes, etc.).

Units were standardized throughout the manuscript

- Lines 227-231: This is unclear. How many residues were used? Previously, three were mentioned, and here, it states ten. Please explain better.

Three wastes were presented in this work and it has been reviewed to reflect that. Thank you

- Lines 227-234: If not all residues are presented, only include the studied material.

Thank you very much, well taken.

- Review the numbering of items, as the article's structure has results before methodology. This applies to the numbering of tables and figures—number according to their order in the text.

The numbering of items was reviewed throughout the manuscript.

- Lines 268-274: Standardize the font size in the sentence.

The font size was standardized. Thank you very much

- Line 276: Change “Figure 1. Figure 1.” to “Figure 1.”

Corrections has been effected on Figure 1. Thank you

- Figure 1: Enlarge the figure and improve the resolution since the caption is unclear.

The figure was enlarged.

- Lines 324-335: Standardize the font size in the sentence.

Thank you very much font size was standardized

- Table 1: Change “Hot trob” to “Hot trub.”

Change effected. Thank you

- Line 341: This term is unclear. What does “Antioxidant sum parameters of extracts” refer to?

The refers to the sum parameters Measured by ABTS and DPPH assay determined the antioxidant capacity of the sum of all chemical species present in the extract. these include polyphenols.reducing sugars and reducing amino acids and peptides

- Figure 2: Improve the resolution of the figure.

Resolution improved

- Figure 3: Adjust and standardize the size of the figure.

The size of figure is standardized and adjusted.

- Lines 396-403: This material and methods should be removed from the results section.

Removed. Thank you

- Lines 434-435: This sentence needs to be reviewed and elaborated on, as it is disconnected now.

The sentence elaborated on. Thank you.

- Line 448: Figure 7B? The sequence of figure numbering is incorrect. The numbering or the way the results are presented must be reviewed.

Figure was repositioned

- Figure 7: Improve the resolution of the figure.

Resolution improved

- Results: The authors should discuss the mechanism of bacterial inhibition.

Results discussed further in terms of mechanism.

- Results: The discussion on the antioxidant mechanism and the role of the compounds in this process should be expanded.

Antioxidant assay was further discussed in the results and the role of compounds

- Antioxidant activity: Specify the reason for the difference concerning the methods' principles and each antioxidant's characteristics.

The reasons for the difference were further discussed

Round 2

Reviewer 3 Report

Comments and Suggestions for Authors

The authors have satisfactorily responded to all my questions and made the necessary changes to the manuscript.